# L-SR1: Learned Symmetric-Rank-One Preconditioning

## Abstract

End-to-end deep learning has achieved impressive results but remains limited by its reliance on large labeled datasets, poor generalization to unseen scenarios, and growing computational demands. In contrast, classical optimization methods are data-efficient and lightweight but often suffer from slow convergence. While learned optimizers offer a promising fusion of both worlds, most focus on first-order methods, leaving learned second-order approaches largely unexplored.

We propose a novel learned second-order optimizer that introduces a trainable pre-conditioning unit to enhance the classical Symmetric-Rank-One (SR1) algorithm. This unit generates data-driven vectors used to construct positive semi-definite rank-one matrices, aligned with the secant constraint via a learned projection. Our method is evaluated through analytic experiments and on the real-world task of Monocular Human Mesh Recovery (HMR), where it outperforms existing learned optimization-based approaches. On the HMR task, it surpasses a fully-trained baseline using only 10% of the training data, underscoring its data efficiency. Featuring a lightweight model and requiring no annotated data or fine-tuning, our method offers strong generalization and is well-suited for integration into broader optimization-based frameworks.

## 1 Introduction

End-to-end deep learning has demonstrated significant power but is constrained by its reliance on large labeled datasets and limited ability to generalize to unseen scenarios. Furthermore, increased model sizes featured in recent works pose a limitation as they demand high compute and memory resources. In contrast, classical optimization excels in data-scarce settings and features a low memory stamp, but often suffers from long runtime due to its iterative nature. To address this, extensive research has focused on accelerating convergence, with optimization methods broadly categorized into first-order and second-order approaches.

First-order methods, such as Adam (Kingma & Ba, 2014) and Nesterov Accelerated Gradient (NAG) (Nesterov, 1983; Sutskever et al., 2013), rely on estimated gradient momentums for parameter updates. Second-order methods, such as Symmetric-Rank-One (SR1) (Conn et al., 1991) and Broyden-Fletcher-Goldfarb-Shanno (BFGS) (Liu & Nocedal, 1989), utilize approximations of the inverse Hessian matrix (Boyd et al., 2004). While more computationally intensive, second-order methods typically achieve faster convergence by accurately capturing the underlying structure of the objective function and exploiting dependencies between variables.

Learned optimization has recently emerged as a promising field that leverages deep learning to enhance traditional optimization methods. These approaches incorporate trainable deep neural network (DNN) architectures—such as Multi-Layer Perceptrons (MLPs) (Andrychowicz et al., 2016; Li & Malik, 2016; Song et al., 2020), Recurrent Neural Networks (RNNs) (Andrychowicz et al., 2016), Transformers (Gärtner et al., 2023), and hybrid models (Metz et al., 2020)—into optimization frameworks. Once trained on specific objectives, these learned optimizers exhibit significantly faster convergence. Despite the increasing popularity of learned optimizers, the integration of learnable components with second-order methods still remains largely unexplored.

By accelerating convergence, learned optimizers offer a compelling path to bridging classical optimization techniques with modern deep learning-based approaches in computer vision tasks.

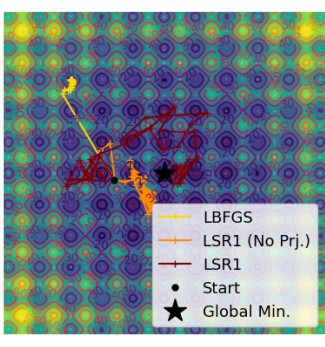 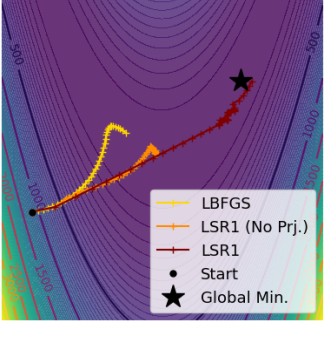 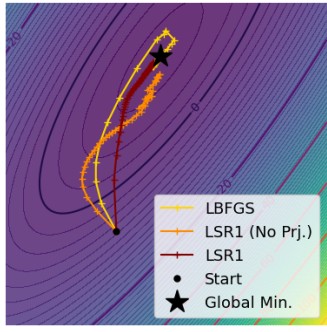

(a) **Rastrigin function.**    (b) **Rosenbrock function.**    (c) **Quadratic function.**

Figure 1: **Optimization trajectories.** Our evaluation spans both classic analytic functions and the real-world human mesh recovery (HMR). Shown here are example optimization trajectories on a quadratic function and two well-known challenging benchmark functions (Surjanovic & Bingham)—the Rosenbrock and Rastrigin functions. In this example, we compare LBFGS with our lightweight L-SR1 method, with and without the proposed learned projection. The learned projection, a novel element of our approach, improves convergence while preserving model compactness.

Monocular Human Mesh Recovery (HMR) seeks to estimate a 3D human mesh from a single 2D image—a fundamentally ill-posed problem due to the inherent loss of depth information. Historically, in the absence of large-scale annotated datasets, optimization-based methods dominated the field (Bogo et al., 2016; Pavlakos et al., 2019). However, with the emergence of increasingly comprehensive annotated datasets, these traditional approaches were largely supplanted by deep neural network (DNN)-based methods. These newer methods span both end-to-end regression models and iterative frameworks (Sun et al., 2023; Shin et al., 2024; Wang et al., 2025), including learned optimization (Kolotouros et al., 2019; Song et al., 2020), achieving state-of-the-art performance. Nonetheless, they come with notable trade-offs, including significantly larger model sizes and a continued reliance on vast amounts of annotated training data.

In this work, we introduce Learned-SR1 (L-SR1), a novel learned second-order optimizer that extends the classical SR1 algorithm. Our method incorporates a learnable module that generates data-driven vectors to approximate the Hessian matrix, enabling more informed and efficient updates. To ensure robustness and theoretical soundness, we also propose a learned projection operation that maintains the positive semi-definiteness of the approximation while satisfying the secant condition—an essential principle of Quasi-Newton methods. By building on the SR1 framework, L-SR1 achieves state-of-the-art performance with a notably compact model size.

We evaluate our approach on both analytic benchmark functions and the real-world task of HMR (Fig. 1). The results show that L-SR1 converges rapidly and efficiently, while also demonstrating strong generalization. Notably, even when trained in a self-supervised manner on limited fractions of the data and without explicit fine-tuning, L-SR1 outperforms existing optimization-based HMR frameworks. These findings highlight the potential of L-SR1 to be seamlessly integrated into broader optimization-based pipelines, significantly enhancing their performance.

We summarize our key contributions as follows:

- We propose Learned-SR1 (L-SR1), a lightweight, self-supervised learned optimizer that integrates a trainable preconditioning unit into the SR1 framework, enabling data-driven curvature estimation without the need for annotated data or supervised meta-training.

- We introduce a learned projection mechanism that enforces both the secant condition and positive semi-definiteness, preserving core Quasi-Newton properties within a learned architecture.

- We demonstrate that L-SR1 can be seamlessly integrated into optimization-based HMR pipelines, where it consistently outperforms both traditional and learned solvers in terms of convergence and generalization.

## 2 RELATED WORK

**Second-Order Optimization and Preconditioning**   Second-order optimization methods leverage curvature information to accelerate convergence, typically by using the inverse Hessian matrix (Bertsekas, 1999). Since computing and inverting the full Hessian is often impractical, Quasi-Newton methods such as DFP and BFGS (Bertsekas, 1999), LBFGS (Nocedal, 1980; Liu & Nocedal, 1989), and SR1 (Conn et al., 1991; Khalfan et al., 1993) were introduced to estimate the inverse Hessian iteratively. SR1, in particular, used rank-one updates and demonstrated favorable convergence under mild conditions. These ideas were extended to large-scale problems in deep learning using scalable approximations (Martens & Grosse, 2015; Gupta et al., 2018; Yao et al.).

**Learned Optimization and Model-Based Deep Learning**   Learned optimization frameworks integrate trainable modules into iterative solvers, generating adaptive update rules from data (Andrychow-icz et al., 2016; Li & Malik, 2016; Metz et al., 2020; Wichrowska et al., 2017). While earlier works focused on first-order dynamics, more recent methods incorporated richer structures, such as Transformers (Gärtner et al., 2023) and low-rank attention mechanisms (Jain et al., 2023). Concurrently, model-based deep learning embedded learning into principled algorithmic formulations to preserve interpretability and robustness (Shlezinger et al., 2020; Revach et al., 2022; Shlezinger et al., 2023).

Within this hybrid space, several methods explored second-order-inspired learned optimizers. Li et al. and Gärtner et al. (2023) proposed learned optimizers inspired by Quasi-Newton principles, while Ayad et al. (2024) constructed an unrolled BFGS-like network for CT reconstruction. These works demonstrated the potential of combining learned and second-order methods, but often relied on large models or explicit supervision. Our approach advances this line by introducing a lightweight, self-supervised SR1-inspired preconditioning unit that enforces the secant condition and improves convergence efficiency and generalization.

**Human Mesh Recovery (HMR)**   Human Mesh Recovery (HMR) aims to estimate 3D human meshes from single RGB images, a highly ill-posed problem due to the loss of depth. Early works approached HMR through optimization-based fitting (Bogo et al., 2016; Pavlakos et al., 2019), which, while data-efficient, were slow and sensitive to initialization. With the emergence of large-scale datasets, deep learning methods were introduced, including regression-based approaches (Shin et al., 2024; Sun et al., 2023; Wang et al., 2025) and iterative refinement techniques, including learned optimization (Kolotouros et al., 2019; Song et al., 2020; Shetty et al., 2023). These models achieved impressive performance but depended heavily on annotated data and large architectures. In contrast, our method integrates a learned SR1-inspired optimizer into the HMR process, outperforming learned optimization-based methods while requiring neither large models nor explicit fine-tuning.

## 3 THEORETICAL BACKGROUND AND PRELIMINARIES

### 3.1 LEARNED OPTIMIZATION

A typical learned optimization framework consists of the following update rule:

$$\mathbf{x}_{k+1} \leftarrow \mathbf{x}_k + \varphi_\Theta\left(\nabla f(\mathbf{x}_k), \mathbf{x}_k, \ldots\right), \tag{1}$$

where $\varphi_\Theta(\cdot)$ is a learnable function parameterized by $\Theta$, which can be conditioned on a variety of features, such as the current iterate and gradient.

Training a learned optimizer, referred to as *meta-training*, involves alternating between *inner* and *outer* iterations. In each outer iteration, the optimizer performs $K$ unrolled inner optimization steps. The *inner* objective values along the optimization trajectory are recorded and used to compute a *meta-loss*, often of the form:

$$\mathcal{L}_{\text{meta}} = \sum_{k=1}^{K} f(\mathbf{x}_k). \tag{2}$$

Gradients of this meta-loss with respect to $\Theta$ are then backpropagated through the unrolled computation graph, and the parameters are updated using a *meta-optimizer*.

## 3.2 QUASI-NEWTON METHODS

Let $f : \mathbb{R}^n \to \mathbb{R}$ be a twice differentiable objective function. The Quasi-Newton (QN) update step is given by

$$\mathbf{x}_{k+1} \leftarrow \mathbf{x}_k - \alpha_k \mathbf{B}_k \mathbf{g}_k, \tag{3}$$

where $\mathbf{B}_k$ is a preconditioning matrix approximating the inverse Hessian at $\mathbf{x}_k$, $\mathbf{g}_k = \nabla f(\mathbf{x}_k)$ is the gradient, and $\alpha_k$ is a step size.

QN methods differ in how they update $\mathbf{B}_k$, but they all satisfy the *secant constraint*, derived from a first-order Taylor approximation. Defining $\mathbf{p}_k = \mathbf{x}_k - \mathbf{x}_{k-1}$ and $\mathbf{q}_k = \mathbf{g}_k - \mathbf{g}_{k-1}$, the secant condition is

$$\mathbf{B}_k \mathbf{q}_k = \mathbf{p}_k. \tag{4}$$

This condition ensures that the preconditioner $\mathbf{B}_k$ captures local curvature information. Additionally, $\mathbf{B}_k$ is typically required to be positive semi-definite to guarantee that the update direction is a descent direction.

# 4 LEARNED SYMMETRIC-RANK-ONE (L-SR1)

L-SR1 is a learned extension of the classical Symmetric Rank-One (SR1) Quasi-Newton method, designed to integrate lightweight trainable modules into a principled second-order optimization framework. The method enhances convergence while maintaining scalability and generalization across problem dimensions.

At its core, SR1 approximates the inverse Hessian matrix using a rank-one update of the form

$$\mathbf{B}_{k+1} \leftarrow \mathbf{B}_k + \mathbf{u}_k \mathbf{v}_k^\top, \tag{5}$$

where the vectors $\mathbf{v}_k = \mathbf{p}_k - \mathbf{B}_k \mathbf{q}_k$ and $\mathbf{u}_k = \mathbf{v}_k / (\mathbf{v}_k^\top \mathbf{q}_k)$ are chosen to satisfy the secant constraint presented in Eq. (4). This ensures that the update captures local curvature information of the objective function.

A significant advantage of the SR1 structure is that it relies solely on outer products of low-dimensional vectors. This allows for a *limited-memory implementation*, where instead of storing the full matrix $\mathbf{B}_k$, L-SR1 maintains a fixed-size *buffer* $\mathcal{B}_L$ containing the most recent $L$ vectors. These vectors are used to reconstruct $\mathbf{B}_k$ implicitly during optimization. If the buffer exceeds its capacity, the oldest entries are discarded, ensuring memory efficiency in high-dimensional problems.

A central design principle in L-SR1 is *invariance to the problem dimension*. All learnable components are constructed to operate element-wise, ensuring that the optimizer generalizes across problem sizes without re-training.

However, a known limitation of SR1 is that its updates do not guarantee positive semi-definiteness of $\mathbf{B}_k$, which can result in non-descent directions and instability. In the learned optimization realm, a naive fix is to use outer products of the form $\mathbf{v}\mathbf{v}^\top$ as was done in (Gärtner et al., 2023), which are always positive semi-definite and symmetric, but such updates fail to satisfy the secant constraint. Traditional methods like LBFGS address this using more elaborate update strategies. In contrast, L-SR1 proposes a vector generator unit accompanied by a *novel learned projection* to efficiently maintain both positive definiteness and compliance with the secant constraint.

A summary of the proposed method is given in Alg. 1 and a block diagram is in Fig 2. We now describe the components of the L-SR1 algorithm in detail. The trainable modules are introduced in Section 4.1, and our proposed learned projection mechanism is described in Section 4.2.

## 4.1 LEARNED COMPONENTS

To enrich SR1 with data-driven flexibility, L-SR1 integrates three neural modules. All modules follow a standard and lightweight multi-layer perceptron (MLP) architecture and operate elementwise over the input dimensions, ensuring compatibility with varying problem sizes. Model architectures are presented in the supplementary material.

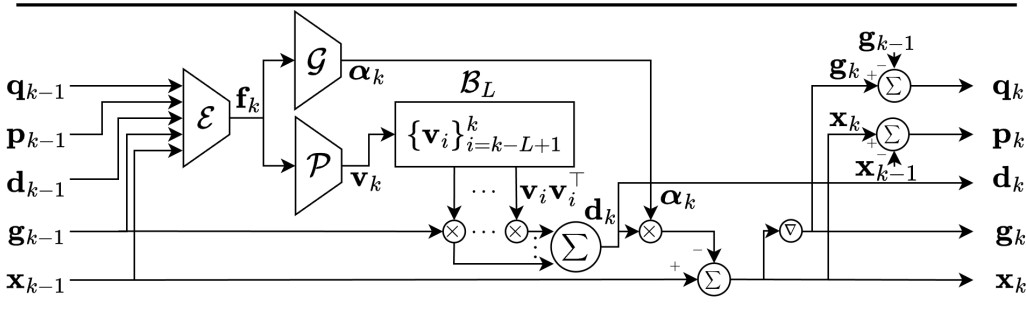

Figure 2: **Learned-SR1 (L-SR1) iteration block diagram.** At each iteration $k$, the Input Encoder $\mathcal{E}$ receives the vectors $\mathbf{x}_{k-1}$, $\mathbf{p}_{k-1}$, $\mathbf{d}_{k-1}$, $\mathbf{g}_{k-1}$, and $\mathbf{q}_{k-1}$, producing a feature vector $\mathbf{f}_k$. This is passed to the Vector Generator $\mathcal{P}$, which outputs a new direction vector $\mathbf{v}_k$, and to the Learning Rate Generator $\mathcal{G}$, which produces element-wise learning rates $\boldsymbol{\alpha}_k$ (Sec. 4.1)). The updated descent direction $\mathbf{d}_k$ is computed as a sum of rank-one terms $\mathbf{v}_i \mathbf{v}_i^\top \mathbf{g}_{k-1}$, using the last $L$ vectors stored in the buffer $\mathcal{B}_L$. Finally, the optimization step is performed using $\boldsymbol{\alpha}_k$ and $\mathbf{d}_k$.

---

**Algorithm 1** Learned-SR1 (L-SR1)

---

**Inputs:** Objective $f \in \mathcal{C}^2$, initial point $\mathbf{x}_0 \in \mathbb{R}^n$, initial gradient $\mathbf{g}_0$, buffer $\mathcal{B}_L = \{\emptyset\}$
 1: Initialize $\mathbf{p}_0 \leftarrow \mathbf{x}_0, \mathbf{q}_0 \leftarrow \mathbf{g}_0, \mathbf{d}_0 \leftarrow \mathbf{g}_0$
 2: **for** $k = 1, 2, \dots$ **until convergence do**
 3:     $\mathbf{f}_k \leftarrow \mathcal{E}(\mathbf{x}_{k-1}, \mathbf{p}_{k-1}, \mathbf{d}_{k-1}, \mathbf{g}_{k-1}, \mathbf{q}_{k-1})$                ▷ Input features
 4:     $\mathbf{v}_k \leftarrow \mathcal{P}(\mathbf{f}_k), \quad \boldsymbol{\alpha}_k \leftarrow \mathcal{G}(\mathbf{f}_k)$                        ▷ Get update
 5:     **if** $|\mathcal{B}_L| = L$ **then**
 6:         Discard oldest element in $\mathcal{B}_L$
 7:     **end if**
 8:     $\mathcal{B}_L \leftarrow \mathcal{B}_L \cup \mathbf{v}_k$                                  ▷ Update buffer
 9:     $\mathbf{d}_k \leftarrow \sum_{\mathbf{v} \in \mathcal{B}_L} \mathbf{v}\mathbf{v}^\top \mathbf{g}_{k-1}$          ▷ Compute descent direction
10:     $\mathbf{x}_k \leftarrow \mathbf{x}_{k-1} - \boldsymbol{\alpha}_k \odot \mathbf{d}_k$                  ▷ Optimization step
11:     $\mathbf{g}_k \leftarrow \nabla f(\mathbf{x}_k), \quad \mathbf{p}_k \leftarrow \mathbf{x}_k - \mathbf{x}_{k-1}, \quad \mathbf{q}_k \leftarrow \mathbf{g}_k - \mathbf{g}_{k-1}$
12: **end for**
**Output:** $\mathbf{x}^* \leftarrow \mathbf{x}_k$

---

**Input Encoder** $\mathcal{E}$    The encoder constructs a latent representation of the optimization state. It takes as input the current point $\mathbf{x}_{k-1}$, previous step $\mathbf{p}_{k-1}$, previous descent direction $\mathbf{d}_{k-1}$, previous gradient $\mathbf{g}_{k-1}$, and previous gradient step $\mathbf{q}_{k-1}$, which are concatenated into an array of shape $N \times 5$, where $N$ is the problem dimension. The encoder then maps this input to a latent representation $\mathbf{f}_k \in \mathbb{R}^{N \times M}$, with $M > 5$, enabling a richer feature space that informs the subsequent modules.

**Vector Generator** $\mathcal{P}$    This module produces a single vector $\mathbf{v}_k \in \mathbb{R}^N$ given $\mathbf{f}_k$ at each iteration. The outer product $\mathbf{v}_k \mathbf{v}_k^\top$ is then used—together with the learned projection described in Section 4.2—to construct the curvature matrix $\mathbf{B}_k$ in a manner analogous to the SR1 update.

**Learning Rate Generator** $\mathcal{G}$    The learning rate generator takes the latent representation from the encoder $\mathbf{f}_k$ and outputs a vector $\tilde{\boldsymbol{\alpha}}_k \in \mathbb{R}^N$, interpreted as the logarithm of coordinate-wise learning rates. The final learning rate vector $\boldsymbol{\alpha}_k$ is computed element-wise using the transformation:

$$\boldsymbol{\alpha}_k = \gamma_1 \cdot \exp\left(\gamma_2 \cdot \tilde{\boldsymbol{\alpha}}_k\right),$$

where $\gamma_1$ and $\gamma_2$ are scalar hyperparameters.

## 4.2   LEARNED PROJECTION

As discussed earlier, our goal is to construct preconditioning matrices $\mathbf{B}_k$ that are both positive semi-definite (PSD) and approximately satisfy the secant equation (Eq. (4)). We formalize this

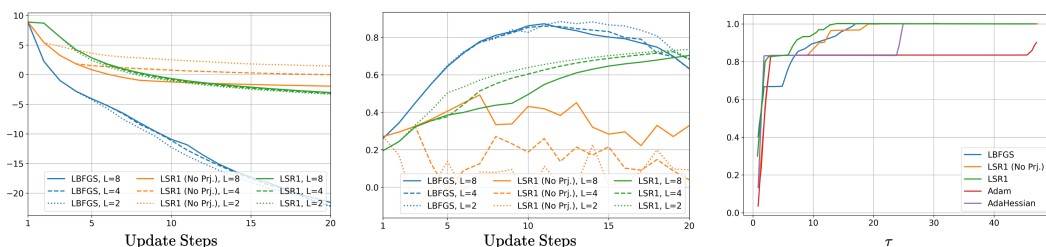

(a) **Quadratic objective values.**  (b) **Cosine similarity to NM.**  (c) **Performance profiles.**

Figure 3: **Analytic optimization experiments.** Figures 3a and 3b show results from our quadratic experiments (Section 5.1.1), comparing L-SR1 (with and without learned projection) to LBFGS (Nocedal, 1980), which serves as a reference. Figures 3a and 3b report objective values and cosine similarities with the Newton direction, respectively, averaged over the test set. The learned projection improves L-SR1's convergence and its alignment with the Newton direction. Figure 3c shows performance profiles on a set of benchmark functions (Section 5.1.2); the high profile indicates the consistent effectiveness of our method.

requirement through a projection objective that minimizes the violation of the secant condition:

$$\mathbf{B}_k^* = \pi_+(\mathbf{B}_k) = \underset{\mathbf{B}_k \in S_+}{\operatorname{argmin}} \|\mathbf{p}_k - \mathbf{B}_k \mathbf{q}_k\|_2^2, \tag{6}$$

where $S_+$ denotes the space of positive semi-definite matrices. This formulation seeks a PSD matrix that best approximates the secant constraint in a least-squares sense.

To ensure scalability and efficient computation, we constrain $\mathbf{B}_k$ to take the following structured form:

$$\tilde{\mathbf{B}}_k = \mathbf{B}_0 + \sum_{i=1}^{L} \mathbf{v}_i \mathbf{v}_i^\top, \tag{7}$$

where each vector $\mathbf{v}_i \in \mathbb{R}^n$ is produced by the vector generator module $\mathcal{P}$ and stored in a fixed-size buffer of length $L$. This construction ensures that $\tilde{\mathbf{B}}_k$ remains positive semi-definite as long as $\mathbf{B}_0 \succ 0$, which we initialize as the identity matrix. The use of outer products is inspired by the classical SR1 update structure and is also consistent with techniques adopted in prior learned optimization works, such as (Gärtner et al., 2023), which promote symmetry and positivity.

The projection is thus formulated as the following optimization problem:

$$\tilde{\mathbf{B}}_k^* = \tilde{\pi}_+(\tilde{\mathbf{B}}_k) = \underset{\tilde{\mathbf{B}}_k = \mathbf{B}_0 + \sum_{i=1}^{L} \mathbf{v}_i \mathbf{v}_i^\top}{\operatorname{argmin}} \|\mathbf{p}_k - \tilde{\mathbf{B}}_k \mathbf{q}_k\|_2^2. \tag{8}$$

We define this objective as the *secant penalty* $\mathcal{R}_{\text{sec}}$, which we incorporate into the overall meta-training loss:

$$\mathcal{L}_{\text{meta}} = \frac{1}{K} \sum_{k=1}^{K} \left( f(\mathbf{x}_k) + \lambda_{\text{sec}} \cdot \|\mathbf{p}_k - \tilde{\mathbf{B}}_k \mathbf{q}_k\|_2^2 \right), \tag{9}$$

where $\lambda_{\text{sec}}$ is a hyperparameter controlling the importance of satisfying the secant condition, and $K$ is the number of unrolled optimization steps per meta-iteration.

This formulation ensures that the learned preconditioning matrices remain PSD and are trained to satisfy the secant constraint effectively, using only the buffer of learned vectors generated at each step. Notably, this does not add computational overhead during inference.

## 5 EXPERIMENTATION

We conduct a series of experiments to evaluate the effectiveness and generalization capabilities of the proposed L-SR1 optimizer. Our evaluation begins with controlled analytic settings: we first isolate the impact of the learned projection mechanism on randomly generated quadratic functions

(Section 5.1.1), then assess generalization across dimensions using a set of benchmark functions (Surjanovic & Bingham) (Section Section 5.1.2). Finally, we demonstrate L-SR1's applicability to real-world problems through a 3D human mesh recovery (HMR) task (Section 5.2), showcasing performance in high-dimensional, structured domains. In each experiment, we compare our results against a range of baselines, which are detailed in the corresponding experimental subsections.

**Implementation Details.** Our method is implemented entirely in PyTorch (Paszke et al., 2019)[1]. Following standard learned optimization practices, our training setup consists of inner optimization loops and outer meta-iterations. Using PyTorch's autograd framework, we compute gradients of the inner objective with respect to the optimization variables during each unrolled step, and gradients of the meta-loss (Eq. (9)) with respect to the optimizer's parameters during each meta-iteration. We use the AdamW (Loshchilov & Hutter, 2017) optimizer for meta-training, with a fixed learning rate of $10^{-4}$, momentum parameters $\beta_1 = 0.9$, $\beta_2 = 0.999$, and weight decay coefficient $\lambda = 0.01$. Unless otherwise stated, meta-training is run for 10K iterations and takes approximately 15 hours on a single NVIDIA GeForce RTX 3090 GPU. Compute runtime and memory analysis is given in the supplementary material.

## 5.1 ANALYTIC EXPERIMENTS

### 5.1.1 QUADRATIC FUNCTIONS

We first evaluate our method on randomly generated quadratic functions of the form

$$f(\mathbf{x}) = \frac{1}{2}\mathbf{x}^\top \mathbf{H}\mathbf{x} + \mathbf{b}^\top \mathbf{x}, \tag{10}$$

where $\mathbf{H} \in \mathbb{R}^{N \times N}$ is a random positive semi-definite matrix and $\mathbf{b} \in \mathbb{R}^N$ is a random vector. A fixed validation set of 32 such functions and corresponding initial points $\mathbf{x}_0$ is used throughout meta-training. At each meta-iteration, random training batches of size 128 are generated, while validation is performed on the fixed set. Both training and validation are performed on functions with $N = 2$. To assess generalization across dimensions, a test set of 32 new functions and initials with $N = 10$ is used. Training is conducted with buffer size $L = 8$, while testing is done with varying buffer sizes. We compare our L-SR1 optimizer, with and without the learned projection mechanism, to LBFGS (Nocedal, 1980), which is explicitly suited to quadratic functions and serves as a reference. Further implementation details are provided in the supplementary material.

**Results** Figure 3a presents the average test loss for the first 20 iterations. As expected, L-BFGS, being data-independent and tailored to quadratic problems, achieves the fastest convergence across all tested buffer sizes. Our L-SR1 with the learned projection consistently outperforms its non-projected variant, highlighting the effectiveness of enforcing the secant constraint. Notably, our method retains strong performance even with reduced buffer sizes, similar to L-BFGS.

Figure 3b shows the average cosine similarity between the descent directions and the corresponding Newton directions on the test set. The version with the projection exhibits steadily increasing similarity, indicating its ability to learn curvature-aware updates. In contrast, the non-projected variant shows less consistent alignment, further underscoring the benefit of our projection mechanism.

### 5.1.2 PERFORMANCE PROFILES

We compare our learned optimizer against several baselines using performance profiles, a standard evaluation method for optimization algorithms (Dolan & Moré, 2002; Sergeyev & Kvasov, 2015; Beiranvand et al., 2017; Gärtner et al., 2023). Let $P$ represent the set of problems and $S$ the set of solvers. For each solver $s$ and problem $p$, the performance measure $m_{p,s}$ is defined as: $m_{p,s} = \frac{\|\hat{\mathbf{x}}_{p,s} - \mathbf{x}^*\|_2}{\|\mathbf{x}_w - \mathbf{x}^*\|_2}$, where $\hat{\mathbf{x}}_{p,s}$ is the solution found by solver $s$ on problem $p$ after $K$ steps, $\mathbf{x}_w$ is the worst solution among solvers, and $\mathbf{x}^*$ is the global optimum. The performance ratio for each solver is: $r_{p,s} = \frac{m_{p,s}}{\min(m_{p,s} \,:\, s \in S)}$, with the best solver achieving $r_{p,s} = 1$.

---

[1]Code will be made publicly available upon acceptance.

The performance profile for solver $s$ is then defined as:

$$\rho_s(\tau) = \frac{1}{|P|}\text{size}\left(\{p \in P \ : \ r_{p,s} \le \tau\}\right). \tag{11}$$

This measures the proportion of problems where the performance ratio of solver $s$ is within a factor $\tau$ of the best.

We meta-train our model on random quadratic functions and on Rosenbrock and Rastrigin functions with $N = 100$. The test set includes 30 benchmark problems: quadratics with condition numbers 1, 100, and 1000, and Rosenbrock and Rastrigin functions with dimensions ranging from 50 to 1000. We compare L-SR1 (with and without the learned projection) to L-BFGS (Nocedal, 1980), Adam (Kingma & Ba, 2014), and AdaHessian (Yao et al.). Figure 3c shows that our method achieves the highest performance profile, demonstrating strong overall effectiveness.

## 5.2 Monocular Human Mesh Recovery (HMR)

We integrate our lightweight L-SR1 optimizer into the learned optimization based HMR framework of (Song et al., 2020), which consists of trainable initialization and update modules inspired by gradient descent (LGD). Adding self-supervision loss terms during training replaced the need for exhaustive and highly engineers loss terms, as in (Bogo et al., 2016; Pavlakos et al., 2019). By replacing their update module with our method, we achieve faster convergence and improved accuracy. Notably, our model delivers competitive results even when trained on only fractions of the data, without explicit fine-tuning, and with a significantly smaller model size.

**Training**   Our model follows the training pipeline of (Song et al., 2020), using the AMASS dataset which consists of 20M human meshes (Mahmood et al., 2019), to predict shape and pose parameters $(\boldsymbol{\beta}_{\text{gt}}, \boldsymbol{\theta}_{\text{gt}})$. The SMPL (Loper et al., 2023) body model[2] is used to reconstruct 3D meshes from these parameters, and the 2D joint locations $\mathbf{x}_{\text{gt}}$ are obtained by projection and used as inputs. At each inner-iteration $k$, the optimizer estimates parameters $(\boldsymbol{\beta}_k, \boldsymbol{\theta}_k)$, which are used to generate predicted 3D and 2D joints, $\mathbf{X}_k$ and $\mathbf{x}_k$, respectively. The reconstruction loss function, defined as

$$f_{\text{rec}}\left(\mathbf{x}_k\right) = \|\mathbf{w} \odot (\mathbf{x}_k - \mathbf{x}_{\text{gt}})\|_1, \tag{12}$$

serves as our inner objective, where $\mathbf{w}$ are confidence weights and $\odot$ stands for element-wise multiplication.

Incorporating self-supervision as in (Song et al., 2020) and our proposed secant constraint, our total meta-loss is:

$$\mathcal{L}_{\text{meta}} = \sum_{k=1}^{K} \left(\lambda_{\text{2D}}f\left(\mathbf{x}_k\right) + \lambda_{\text{self}}\|\Theta_k - \Theta_{\text{gt}}\|_1\right) + \lambda_{\text{sec}}\mathcal{R}_{\text{sec}}, \tag{13}$$

where $\Theta_k = \{\mathbf{X}_k, \boldsymbol{\theta}_k, \boldsymbol{\beta}_k\}$ and $\lambda_{\text{2D}}$, $\lambda_{\text{self}}$, and $\lambda_{\text{sec}}$ are hyperparameters. Further implementation details are available in the supplementary material.

**Evaluation on 3DPW**   We evaluate our method on the challenging 3DPW dataset (von Marcard et al., 2018), which features complex, in-the-wild poses. We report PA-MPJPE on the test set, following the evaluation protocol of (Song et al., 2020), and using the 2D keypoints provided by OpenPose (Cao et al., 2019). Table 1 summarizes reported errors and required inner iterations for learned optimization methods. Our L-SR1 method outperforms LGD in accuracy while maintaining a smaller model size. Qualitative examples are shown in Fig. 5, with additional examples available in the supplementary material.

---

[2]Although more sophisticated and accurate body models exist, such as (Pavlakos et al., 2019; Osman et al., 2020), SMPL (Loper et al., 2023) was chosen for its widespread availability and ease of integration.

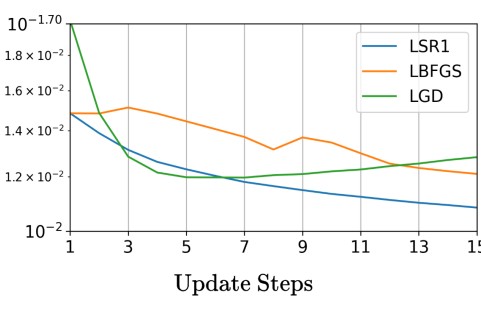

(a) **Reprojection error curves.**

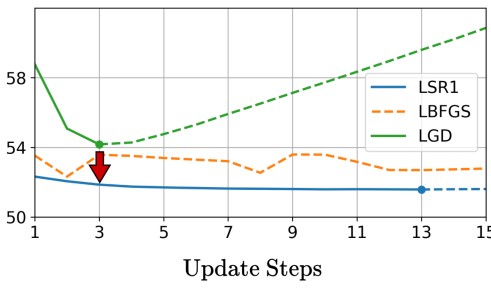

(b) **PA-MPJPE curves.**

Figure 4: **HMR error curves on 3DPW.** We compare 15 inner iterations of LGD (Song et al., 2020), L-SR1 (ours), and LBFGS (Liu & Nocedal, 1989), which is initialized using our trained initialization module. Fig. 4a shows 2D joint reprojection error; Fig. 4b shows PA-MPJPE. Dots mark the suggested stopping steps. While LGD briefly achieves lower 2D error, our method consistently outperforms all others in 3D accuracy and convergence, suggesting it has acquired better internalized 3D priors.

Table 1: **Evaluation on 3DPW.** Official PA-MPJPE of optimizaion based frameworks are given. For learned optimizers, we also report the number of steps (Steps) to reach the lowest error and model size (# Params). For our method, the value in brackets shows the error after 4 steps, for direct comparison with LGD (Song et al., 2020).

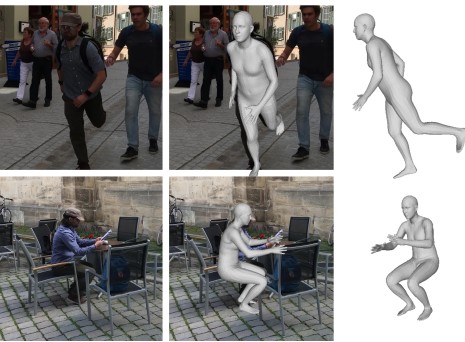

Figure 5: **Qualitative examples from 3DPW.**

| Method | PA-MPJEPE | Steps | # Params |
|---|---|---|---|
| SMPLify (Bogo et al., 2016) | 106.80 | – | – |
| SPIN (Kolotouros et al., 2019) | 59.20 | – | – |
| LGD (Song et al., 2020) | 55.90 | 4 | 17.4 M |
| L-SR1 (Ours) | **51.58** (51.74) | 13 | **10.4 M** |

Table 2: **Data efficiency study.** L-SR1 trained on fractions of AMASS and evaluated on 3DPW.

| Fraction of Data | 80% | 10% | 1% | 0.1% |
|---|---|---|---|---|
| **PA-MPJPE** | 51.67 | 53.03 | 56.14 | 70.78 |

**Data Efficiency Study** To assess generalization beyond the training data, we trained L-SR1 on fractions of the AMASS dataset (80%, 10%, 1% and 0.1%) and evaluated on the full 3DPW test set. Even when trained on only 10% of the data, L-SR1 achieved a test error of 53.03, which still outperforms LGD trained on the full dataset (55.90). With only 1% of the data, the error is 56.14. These results, summarized in Table 2, demonstrate the robustness and data efficiency of our method.

## 6 CONCLUSIONS

We introduced a novel trainable second-order preconditioning unit that enhances the SR1 algorithm through a projection operation, ensuring positive definite preconditioning matrices that satisfy the secant constraint. Our method outperforms existing optimizers in benchmark tasks, achieving the highest performance profile and converging to the global minimum faster, even in challenging problems. In Human Mesh Recovery (HMR), our approach improves convergence speed and accuracy, achieving competitive results with a smaller model size and no explicit fine-tuning.

We believe our method can accelerate gradient-based optimization frameworks, reducing runtime by converging in fewer iterations.

## REPRODUCIBILITY STATEMENT

We provide detailed implementation details in Appendix C and experimental settings in Section 5. Furthermore, the full code used for our experiments is included in the supplementary material to facilitate replication and verification of our results.

## USE OF LARGE LANGUAGE MODELS

We used large language models (LLMs) to assist with writing, phrasing, and simple code tasks during manuscript preparation. All scientific content, technical derivations, experimental design, and data analysis were independently verified and authored by the research team.

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

## A  COMPUTATIONAL ANALYSIS

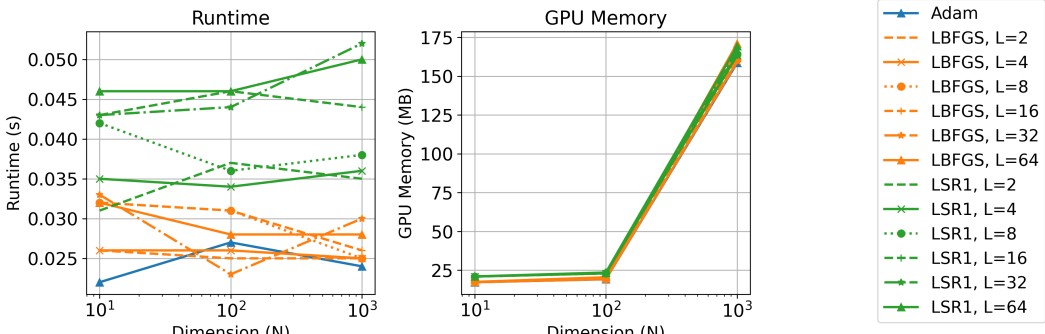

Figure 6: **Computational effort during inference.** Measurements were collected on a single NVIDIA RTX 3090 GPU and correspond to the mean runtime per inner inference iteration and peak memory with a batch size of 32.

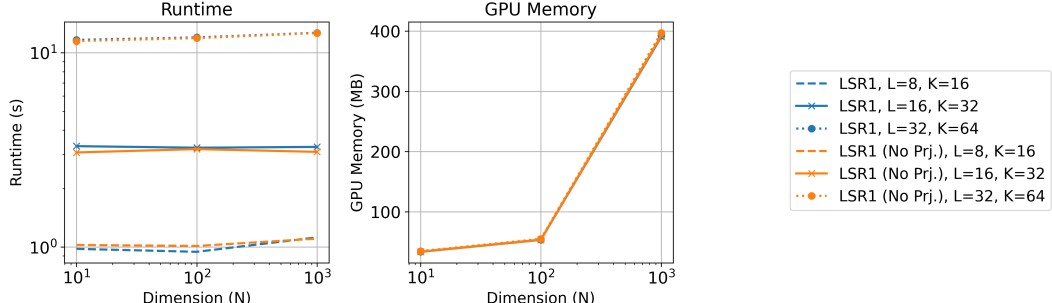

Figure 7: **Computational effort during meta-training.** Measurements were collected on a single NVIDIA RTX 3090 GPU and correspond to the mean runtime per inner inference iteration and peak memory usage with a batch size of 4.

Table 3: **HMR runtime and memory comparison.** Measurements were collected on a single NVIDIA RTX 3090 GPU and correspond to the mean runtime per inner inference iteration and peak memory usage, using a batch size of 256.

| Method | Runtime (ms) | Memory (GiB) |
|---|---|---|
| LGD Song et al. (2020) | 166 | 17.81 |
| L-SR1 | 91 | 14.60 |

## B  ABLATIONS

We conducted a series of ablation studies on our model components. All experiments were carried out on the validation set, which comprises 32 quadratic functions as detailed in Appendix C.2.1. Table 4 summarizes the results. We evaluated the impact of varying the hidden dimension as well as different combinations of inputs to the encoder. The selected configuration is highlighted in bold.

Table 4: **Varying hidden dimension and encoder inputs.** Quadratic validation set errors for different configurations of the hidden dimension and selected inputs to the encoder.

| Hidden dim. | Encoder Inputs | | | | | Valid. loss |
|---|---|---|---|---|---|---|
| $d_{\text{hidden}}$ | $\mathbf{x}_{k-1}$ | $\mathbf{p}_{k-1}$ | $\mathbf{d}_{k-1}$ | $\mathbf{g}_{k-1}$ | $\mathbf{q}_{k-1}$ | |
| 128 | ✓ | ✗ | ✗ | ✓ | ✗ | -2.09 ± 1.85 |
| 128 | ✓ | ✗ | ✓ | ✓ | ✗ | -2.16 ± 1.95 |
| 128 | ✓ | ✓ | ✗ | ✓ | ✓ | -2.51 ± 2.64 |
| 64 | ✓ | ✓ | ✓ | ✓ | ✓ | -1.90 ± 1.76 |
| **128** | ✓ | ✓ | ✓ | ✓ | ✓ | **-2.56 ± 2.80** |

## C  IMPLEMENTATION DETAILS

### C.1  L-SR1 MODULES ARCHITECTURES

Our proposed L-SR1 model comprises three learnable modules: an Input Encoder $\mathcal{E}$, a Vector Generator $\mathcal{P}$, and a Learning Rate (LR) Generator $\mathcal{G}$. All modules operate element-wise and share a common MLP-based architecture, detailed in Tables 5 and 6.

We set $d_{\text{hidden}} = 128$ in all experiments. As described in Section 4.1 and justified in Appendix B, the Input Encoder uses $d_{\text{in}} = 5$ and $d_{\text{out}} = d_{\text{hidden}}$, while both the Vector and LR Generators use $d_{\text{in}} = d_{\text{hidden}}$ and $d_{\text{out}} = 1$.

Table 5: **MLP architecture**

| Layer | Type | Parameters |
|---|---|---|
| fc1 | Linear | Input: $d_{\text{in}}$, Output: $d_{\text{hidden}}$ |
| bn1 | BatchNorm | Features: $d_{\text{hidden}}$ |
| prelu | PReLU | – |
| do1 | Dropout | – |
| MLP1 | Basic Block | Features: $d_{\text{hidden}}$ |
| MLP2 | Basic Block | Features: $d_{\text{hidden}}$ |
| fc2 | Linear | Input: $d_{\text{hidden}}$, Output: $d_{\text{out}}$ |

Table 6: **Basic Block Architecture**

| Layer | Type | Parameters |
|---|---|---|
| fc1 | Linear | Input: $d_{\text{in}}$, Output: $d_{\text{hidden}}$ |
| bn1 | BatchNorm | Features: $d_{\text{hidden}}$ |
| prelu | PReLU | – |
| do1 | Dropout | – |
| fc2 | Linear | Input: $d_{\text{hidden}}$, Output: $d_{\text{hidden}}$ |
| bn2 | BatchNorm | Features: $d_{\text{hidden}}$ |
| do2 | Dropout | – |

### C.2  EXPERIMENTAL SETUP

#### C.2.1  QUADRATIC FUNCTIONS

**Data**  We generate random quadratic functions of the form

$$f(\mathbf{x}) = \frac{1}{2}\mathbf{x}^\top \mathbf{H}\mathbf{x} + \mathbf{b}^\top \mathbf{x}, \tag{14}$$

where $\mathbf{H} \in \mathbb{R}^{N \times N}$ is a positive semi-definite matrix and $\mathbf{b} \in \mathbb{R}^N$ is a random vector.

To construct $\mathbf{H}$, we draw a matrix $\mathbf{A} \in \mathbb{R}^{N \times N}$ from a standard normal distribution and define

$$\mathbf{H} = \mathbf{A}^\top \mathbf{A}, \tag{15}$$

ensuring positive semi-definiteness. We compute the condition number of each $\mathbf{H}$ and discard those exceeding 1000. This process is repeated until full validation and test bathces are acquired. Each matrix is then normalized to have unit Frobenius norm. Independently, $\mathbf{b}$ is sampled from a standard normal distribution and normalized to have unit Euclidean norm.

During training batch generation, we relax the condition number constraint and accept all generated matrices, regardless of their conditioning. We use $N = 2$ for both training and validation, and $N = 10$ for testing. Training batches consist of 128 samples, while validation and test sets each contain 32 samples.

**Hyperparameters** We use the AdamW optimizer (Loshchilov & Hutter, 2017) for meta-training, with a fixed learning rate of $10^{-4}$, momentum parameters $\beta_1 = 0.9$ and $\beta_2 = 0.999$, and a weight decay coefficient of $\lambda = 0.01$. Training is conducted for 10,000 meta-iterations.

We set the buffer size to $L = 8$ and the number of unrolled iterations to $K = 16$. The Learning Rate Generator is configured with scaling parameters $\gamma_1 = 0.4$ and $\gamma = 0.001$. A secant constraint is applied with a weighting factor of $\lambda_{\text{sec}} = 100$. The learning rates for non-trainable optimizers were selected through hyperparameter tuning.

### C.2.2 PERFORMANCE PROFILES

**Data** The quadratic functions used in this experiment are a subset of those defined in Eq. 14, where $\mathbf{b} = \mathbf{0}$ and $\mathbf{H}$ is constrained to be diagonal. Our objective set comprises four such quadratic functions with condition numbers 1, 100, 1000, and 10000, along with the Rosenbrock and Rastrigin functions (Surjanovic & Bingham). Each function is evaluated at input dimensions $N = 50, 100, 250, 500,$ and 1000, yielding a total of 30 distinct optimization problems.

The solver set includes our proposed L-SR1 optimizer, both with and without the learned projection, along with three non-trainable baselines: L-BFGS (Nocedal, 1980), Adam (Kingma & Ba, 2014), and AdaHessian (Yao et al.), totaling six solvers. The learning rates for non-trainable optimizers were selected through hyperparameter tuning.

Each trainable optimizer is meta-trained on three distinct tasks: a randomly generated quadratic function (as defined in Eq. 14), the Rosenbrock function, and the Rastrigin function. For each task, a validation set of 8 fixed initial points is created and remains unchanged throughout meta-training. During each meta-iteration, a new training batch of 8 initial points is generated. Both training and validation are performed with $N = 100$.

Table 7: **Hyperparameters used in Performance profiles**

| Parameter | Quadratics | | Rosenbrock | | Rastrigin | |
|---|---|---|---|---|---|---|
| | Train | Test | Train | Test | Train | Test |
| LR gen. param. $\gamma_1$ | 0.1 | 0.1 | 0.1 | 0.1 | 0.1 | 0.1 |
| LR gen. param. $\gamma_2$ | 0.001 | 0.001 | 0.001 | 0.001 | 0.001 | 0.001 |
| Buffer size $L$ | 16 | 64 | 32 | 64 | 32 | 64 |
| Unrolled iterations $K$ | 32 | - | 64 | - | 64 | - |
| Secant loss weight $\lambda_{\text{sec}}$ | 10 | - | 1 | - | 1 | - |

**Hyperparameters** We use the AdamW optimizer (Loshchilov & Hutter, 2017) for meta-training, with a fixed learning rate of $10^{-4}$, momentum parameters $\beta_1 = 0.9$ and $\beta_2 = 0.999$, and a weight decay coefficient of $\lambda = 0.01$. Training is conducted for 10,000 meta-iterations. Used hyperparameters are summarized in Table 7.

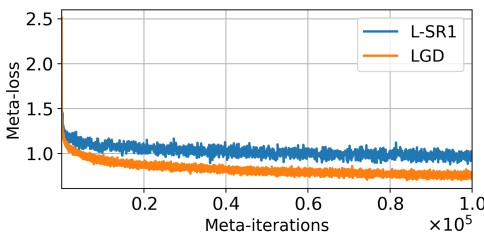 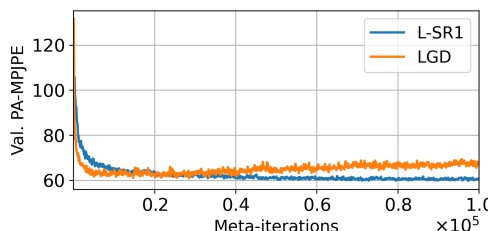

(a) **HMR meta-loss on AMASS Mahmood et al. (2019).**

(b) **HMR validation error on 3DPW von Marcard et al. (2018).**

Figure 8: **HMR meta-training on AMASS and validation erros on 3DPW.** Shown first 100K iterations. Our meta-loss is higher as it has the secant loss added to it.

### C.2.3 MONOCULAR HUMAN MESH RECOVERY (HMR)

**Data** We follow the training and evaluation protocol of (Song et al., 2020). Training is performed on the AMASS dataset (Mahmood et al., 2019)[3], which comprises SMPL body models (Loper et al., 2023) parameterized by $(\boldsymbol{\beta}_{\text{gt}}, \boldsymbol{\theta}_{\text{gt}})$. At each training iteration, a batch of 2D joints is generated by projecting the corresponding 3D joints onto a randomly sampled camera view.

Validation and testing are conducted on the 3DPW dataset (von Marcard et al., 2018) using the same protocol as (Song et al., 2020), utilizing the provided 2D joint detections obtained via OpenPose (Cao et al., 2019). During training, we evaluate our model on the official 3DPW validation set and retain the checkpoint that achieves the best validation performance. The results reported in the paper are obtained by evaluating this best-performing model on the official 3DPW test set. Meta-training and validation graphs are given in Fig. 8.

**Hyperparameters** We meta-train for 400K iterations using the AdamW optimizer (Loshchilov & Hutter, 2017), with an initial learning rate of $10^{-3}$, which is decayed by a factor of $\gamma = 0.8$ every 20K iterations. The momentum parameters are set to $\beta_1 = 0.9$ and $\beta_2 = 0.999$, and the weight decay coefficient is $\lambda = 0.01$.

We configure the meta-optimization process with a buffer size of $L = 4$ and $K = 8$ unrolled iterations. The Learning Rate Generator is parameterized with scaling factors $\gamma_1 = 0.1$ and $\gamma = 0.001$. The secant constraint is weighted with $\lambda_{\text{sec}} = 1$.

## D LIMITATIONS

Despite its strong performance, our learned second-order optimizer has certain limitations—most notably, a relatively high runtime compared to first-order methods such as Adam (Kingma & Ba, 2014) (see Appendix A). However, in optimization settings where loss evaluation and gradient computation are the primary computational bottlenecks, our method can be effectively integrated to reduce the number of iterations required, thereby offering overall performance gains, as demonstrated in this paper.

---

[3]The AMASS (Mahmood et al., 2019) dataset an aggregation of the following datasets (Advanced Computing Center for the Arts and Design; Helm et al., 2015; Ghorbani et al., 2020; Troje, 2002; Carnegie Mellon University; Aristidou et al., 2019; Bogo et al., 2017; Ltd.; Taheri et al., 2020; Brahmbhatt et al., 2019; Müller et al., 2007; Chatzitofis et al., 2020; Sigal et al., 2010; Mandery et al., 2015; 2016; Krebs et al., 2021; Loper et al., 2014; Tripathi et al., 2023; Akhter & Black, 2015; University & of Singapore; Ghorbani & Black, 2021; Hoyet et al., 2012; Trumble et al., 2017; Li et al., 2024).

# E    HMR QUALITATIVE RESULTS

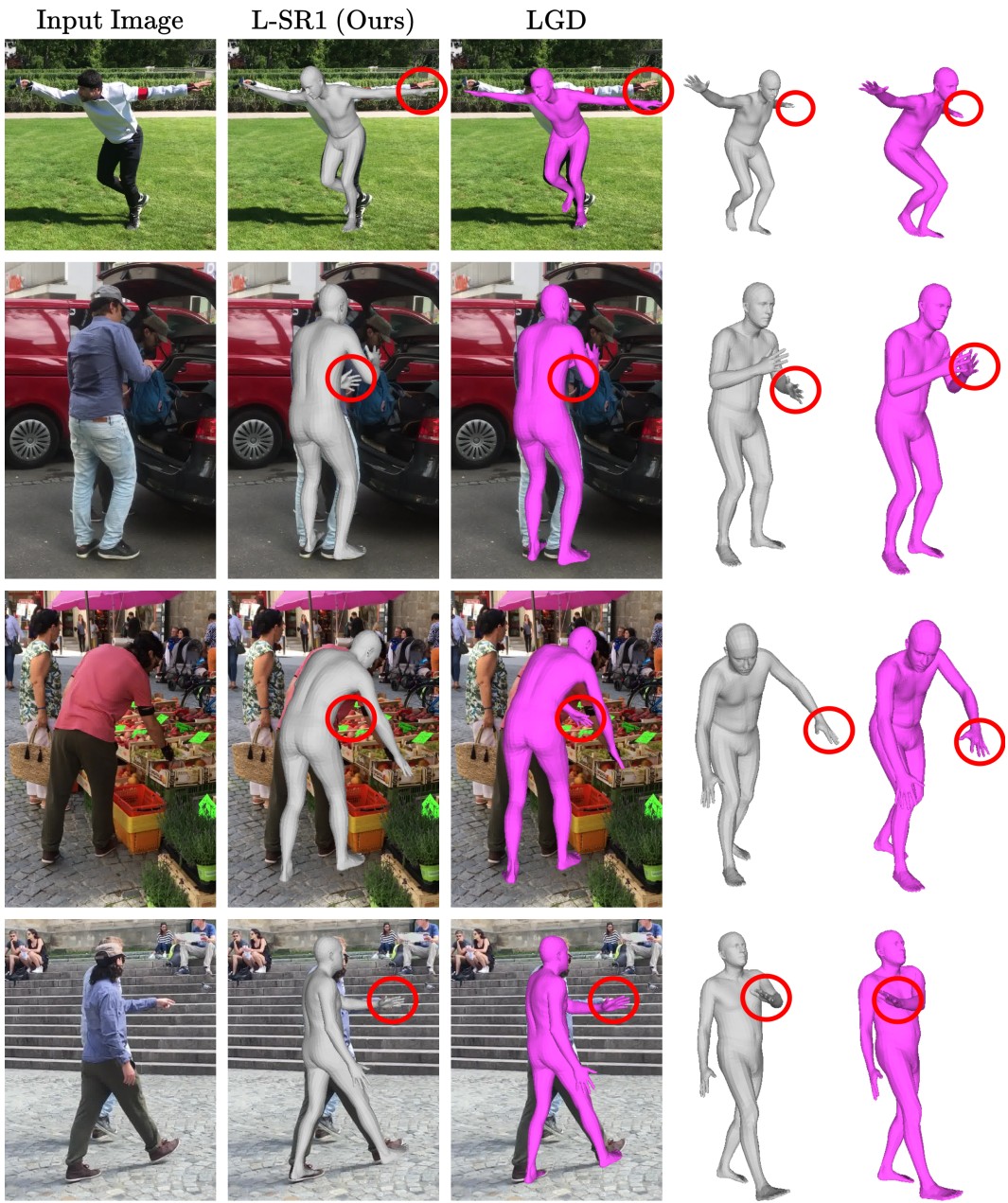

Figure 9: **Qualitative comparison of L-SR1 (ours) and LGD HMR results.** Meshes optimized with L-SR1 (Ours) are shown in white, while those from LGD Song et al. (2020) are shown in pink. Regions of interest are highlighted with red circles.

# F    THEORETICAL BACKGROUND

## F.1    QUASI-NEWTON APPROACH

Consider the Newton Method (NM), which utilizes the full Hessian matrix for preconditioning via directional adjustments. The NM minimization direction is defined as

$$\mathbf{d}_{\text{NM}} = -\mathbf{H}^{-1}(\mathbf{x}_k)\nabla f(\mathbf{x}_k), \tag{16}$$

where $\mathbf{H}^{-1}(\mathbf{x}_k)$ represents the inverse Hessian matrix evaluated at $\mathbf{x}_k$. Notably, when $f$ is quadratic, NM theoretically converges in a single iteration. However, the practicality of NM is often hindered by the challenge of computing and inverting the Hessian matrix. Consequently, a class of second-order optimization methods, termed Quasi-Newton (QN), emerged, aiming to approximate the inverse Hessian matrix, denoted $\mathbf{B}$, during the optimization process. This involves updating $\mathbf{B}$ iteratively alongside each optimization step. Algorithm 2 presents a summary of the QN optimization approach.

## F.2    SYMMETRIC-RANK-ONE (SR1)

One prominent Quasi-Newton (QN) method is the Symmetric-Rank-One (SR1) technique, which involves iteratively accumulating symmetric rank-one matrices to estimate $\mathbf{B}$. Let $\mathbf{g}_k = \nabla f(\mathbf{x}_k)$ and $\mathbf{B}_k = \mathbf{H}^{-1}(\mathbf{x}_k)$ denote the gradient and inverse Hessian at step $k$, respectively. Considering the linear approximation of $\mathbf{g}_{k+1}$ as:

$$\mathbf{g}_{k+1} = \mathbf{g}_k + \mathbf{H}(\mathbf{x}_k)\left(\mathbf{x}_{k+1} - \mathbf{x}_k\right), \tag{17}$$

and defining $\mathbf{p}_k = \mathbf{x}_{k+1} - \mathbf{x}_k$ and $\mathbf{q}_k = \mathbf{g}_{k+1} - \mathbf{g}_{k1}$, equation (17) reduces to:

$$\mathbf{p}_k = \mathbf{B}_k \cdot \mathbf{q}_k. \tag{18}$$

This equation imposes a constraint directly on $\mathbf{B}_k$, known as the "secant constraint".

The SR1 method involves updating $\mathbf{B}$ with rank-one matrices of the form:

$$\mathbf{B}_k \leftarrow \mathbf{B}_{k-1} + \mathbf{u}\mathbf{v}^{\text{Tr}}, \tag{19}$$

where $\mathbf{u}, \mathbf{v} \in \mathbb{R}^N$. Assuming $\mathbf{B}_{k-1}$ is symmetric, $\mathbf{B}_k$ is symmetric as well. Enforcing the secant constraint given in equation (18) on $\mathbf{B}_k$ yields:

$$\mathbf{p}_k = \left(\mathbf{B}_{k-1} + \mathbf{u}\mathbf{v}^{\text{Tr}}\right) \cdot \mathbf{q}_k, \tag{20}$$

which can be rearranged as:

$$\mathbf{u} = \frac{\mathbf{p}_k - \mathbf{B}_{k-1}\mathbf{q}_k}{\mathbf{v}^{\text{Tr}}\mathbf{q}_k}. \tag{21}$$

This implies that for every $\mathbf{v} \not\perp \mathbf{q}_k$, a suitable $\mathbf{u}$ satisfying the secant constraint can be found. A common choice is $\mathbf{v} = \mathbf{p}_k - \mathbf{B}_{k-1}\mathbf{q}_k$, leading to the following SR1 update step:

$$\mathbf{B}_k \leftarrow \mathbf{B}_{k-1} + \frac{\left(\mathbf{p}_k - \mathbf{B}_{k-1}\mathbf{q}_k\right)\left(\mathbf{p}_k - \mathbf{B}_{k-1}\mathbf{q}_k\right)^{\text{Tr}}}{\left(\mathbf{p}_k - \mathbf{B}_{k-1}\mathbf{q}_k\right)^{\text{Tr}}\mathbf{q}_k}. \tag{22}$$

The SR1 'Update B' process, summarized in Algorithm 3, notably ensures the symmetry of the estimated $\mathbf{B}$, a desirable feature. However, it falls short in guaranteeing positivity, a crucially property, as emphasized in the subsequent lemma.

**Lemma 1.** *Let $f \in \mathcal{C}^2$ be an objective function, and $\mathbf{d}_{NM} = -\mathbf{B}\,\mathbf{g}$ represent the Newton direction. Then, $\mathbf{B} \succ \mathbf{0}$ ensures that $\mathbf{d}_{NM}$ is a descent direction.*

---

**Algorithm 2** Quasi-Newton (QN) Optimization

---

**Inputs:**
    Objective function $f \in \mathcal{C}^1$
    Initial point $\mathbf{x}_0 \in \mathbb{R}^n$
1: **procedure** QUASINEWTON($f$,$\mathbf{x}_0$)
2:    Initialize $\mathbf{B}_0 \leftarrow \mathbf{I}$
3:    **for** $k = 1, 2, ...$ **until convergence do**
4:        $\mathbf{d}_k \leftarrow -\mathbf{B}_{k-1}\nabla f(\mathbf{x}_k)$                ▷ Compute descent direction
5:        Choose $\alpha_k$ such that $f(\mathbf{x}_{k-1} + \alpha_k \cdot \mathbf{d}_k) < f(\mathbf{x}_{k-1})$       ▷ Find step size
6:        $\mathbf{x}_k \leftarrow \mathbf{x}_{k-1} + \alpha_k \cdot \mathbf{d}_k$                    ▷ Optimization step
7:        $\mathbf{B}_k \leftarrow$ UPDATE $(\mathbf{B}_{k-1}, \mathbf{x}_k - \mathbf{x}_{k-1}, \nabla f(\mathbf{x}_k) - \nabla f(\mathbf{x}_{k-1}))$   ▷ Update $\mathbf{B}_k$
8:    **end for**
9: **end procedure**
**Output:** $\mathbf{x}^* \leftarrow \mathbf{x}_k$

---

---

**Algorithm 3** Update B (SR1)

---

**Inputs:**
    Inverse hessian estimate $\mathbf{B}_{k-1}$
    Optimization step $\mathbf{p}_k \triangleq \mathbf{x}_{k+1} - \mathbf{x}_k$
    Gradient step $\mathbf{q}_k \triangleq \mathbf{g}_{k+1} - \mathbf{g}_{k1}$
1: **procedure** UPDATEB($\mathbf{B}_{k-1}$,$\mathbf{p}_k$,$\mathbf{q}_k$)
2:    Set $\mathbf{v} \leftarrow \mathbf{p}_k - \mathbf{B}_{k-1}\mathbf{q}_k$
3:    **if** $\mathbf{v} \not\perp \mathbf{q}_k$ **then**
4:        $\mathbf{B}_k \leftarrow \mathbf{B}_{k-1} + \frac{(\mathbf{p}_k - \mathbf{B}_{k-1}\mathbf{q}_k)(\mathbf{p}_k - \mathbf{B}_{k-1}\mathbf{q}_k)^{\mathrm{Tr}}}{(\mathbf{p}_k - \mathbf{B}_{k-1}\mathbf{q}_k)^{\mathrm{Tr}}\mathbf{q}_k}$       ▷ Update $\mathbf{B}$
5:    **else**
6:        $\mathbf{B}_k \leftarrow \mathbf{B}_{k-1}$                      ▷ Do not update $\mathbf{B}$
7:    **end if**
8: **end procedure**
**Output:** $\mathbf{B}_k$

---

*Proof.* $\mathbf{d}_{\mathrm{NM}}$ is a descent direction if and only if the directional derivative of $f$ in the direction $\mathbf{d}_{\mathrm{NM}}$ satisfies $f'_{\mathbf{d}_{\mathrm{NM}}} \leq 0$. Since $f'_{\mathbf{d}} = \mathbf{g}^{\mathrm{Tr}}\mathbf{d}$,

$$0 \geq f'_{\mathbf{d}_{\mathrm{NM}}} = \mathbf{g}^{\mathrm{Tr}}\mathbf{d}_{\mathrm{NM}} = -\mathbf{g}^{\mathrm{Tr}}\mathbf{B}\,\mathbf{g} \Leftrightarrow \mathbf{g}^{\mathrm{Tr}}\mathbf{B}\,\mathbf{g} \geq 0 \tag{23}$$

Hence, $\mathbf{B} \succcurlyeq \mathbf{0}$ is a sufficient condition ensuring $\mathbf{d}_{\mathrm{NM}}$ is a descent direction. $\square$

Consequently, several methods have been proposed to either eliminate 'bad' directions or ensure positive matrices through more sophisticated schemes.

