# OpenReview forum: "L-SR1: Learned Symmetric-Rank-One Preconditioning"
_ICLR.cc/2026/Conference — Submitted to ICLR 2026_

### Official Review · Reviewer_n9Pd · 2025-10-28

**Soundness:** 3
**Presentation:** 3
**Contribution:** 3
**Rating:** 6
**Confidence:** 4

**Summary:**

Based on the SR1 concept, this paper proposes a Learnable Second-Order Precondition (L-SR1) that learns a positive semi-definite low-rank precondition through three lightweight modules: an input encoder, a vector generator, and a coordinate-wise learning rate generator. A secant-line consistency penalty is introduced during meta-training to enforce constraints, without adding any extra computation during inference. The proposed method is verified on analytical benchmark function families and humanoid mesh recovery (HMR, 3DPW). After replacing the update module of LGD in HMR, L-SR1 achieves higher accuracy, fewer steps, and fewer parameters, demonstrating clear advantages in data efficiency and computational and memory cost.

**Strengths:**

1. The paper proposes a trainable preconditioning unit that augments classical SR1 using data-driven rank-one components and a learned projection that enforces secant consistency while keeping the preconditioner positive semi-definite. This represents an uncommon and creative combination in the learned-optimizer space.
2. It reframes the question of how to satisfy the secant constraint without losing positive semi-definiteness as a learnable projection or penalty design, which removes a long-standing instability of SR1 in a way that is amenable to meta-learning.
3. On quadratic functions and classic benchmarks, the method with the learned projection shows faster decrease and better alignment with a high-quality direction. The performance-profile evaluation indicates the best overall profile among baselines, reflecting a solid and standardized methodology.
4. In HMR (3DPW), the approach improves accuracy over a learned gradient descent baseline while using fewer parameters. The reported curves and tables show the steps to the best error and the model size, following known evaluation protocols.
5. The paper clearly explains the SR1 background, introduces the learned components and projection, and then transitions to analytic and real-world experiments with a consistent and coherent narrative.
6. The work addresses a real pain point in learned optimization. Most learned optimizers are still first-order methods, while this approach advances learned second-order optimization with a lightweight and stable design, lowering the barrier to effectively using curvature information in practice.

**Weaknesses:**

1. The main comparisons are with LGD and an LBFGS variant, but the table omits many recent HMR systems and learned optimizers that are well recognized in the community. Please include fair, apples-to-apples comparisons with at least one or two recent strong HMR frameworks (using the same 2D detections and evaluation protocol) and add a modern learned optimizer baseline (e.g., transformer-based) under the same iteration budget.
2. The ablations cover hidden size and encoder inputs, but not the parameters that likely affect stability and generalization, such as buffer length, unroll steps, secant-penalty weight, and learning-rate scaling. Please provide a grid or heatmap over buffer length × unroll steps × penalty weight × LR scale, reporting final error, iterations to target, failure or “bad-step” rate, and variance across random seeds.
3. It is not entirely clear how the buffer, base matrix, and generated vectors interact during inference beyond the training-time penalty. Please add clear, step-by-step pseudocode for inference, along with a table reporting test-time constraint residuals, curvature–direction alignment, and any guardrails used (e.g., fallback or damping).

**Questions:**

1. Are any consistency checks or safeguard mechanisms (such as trimming, damping, or fallback) applied during inference? Please provide a small table reporting the constraint residuals at each test step, the proportion of failed steps, and the frequency of guardrail activations.
2. Please supplement the performance profile using exactly the same number of iterations and/or the same wall-clock budget. Additionally, include a curve showing the time or number of steps required to reach the target error, ensuring that the conclusions are not affected by asymmetric budgets or unbalanced rollout lengths.
3. Please provide a small grid or heatmap varying buffer length, unroll steps, projection penalty weight, and learning-rate scaling, and report the final error, the number of steps needed to reach the target, the proportion of bad steps, and the variance across random seeds.

---

> ### Author Response · Authors · 2025-11-20
>
> We thank the reviewer for the careful reading and constructive feedback. We address the concerns raised and provide answers to the specific questions posed.
>
> 1. Addressing the Reviewer's Concerns:
>
> Lacking Comparison:
>
> Apart from serving as an interesting non-convex, high-dimensional testbed, the goal of the HMR experiment is to evaluate the generalization behaviour of the optimizer itself, following the same meta-training setup as LGD: training on AMASS with synthetically generated 2D keypoints and evaluating on real 3DPW images. Recent strong HMR systems, to the best of our knowledge, do not operate in this optimizer-centric regime—they are large regressors trained and extensively fine-tuned for HMR—so they do not form an apples-to-apples comparison.
>
> Regarding learned optimizers, unfortunately we were unable to identify recent open-source methods applicable to high-dimensional tasks like HMR beyond LGD (e.g., Transformer-based). To partially compensate, we compare L-SR1 against its non-projected variant in the analytic experiments and against LGD in HMR, consistently observing improvements. If the reviewer can point us to a suitable modern learned optimizer with available code, we would make an effort to include it in the revision.
>
> Lacking Ablations:
>
> We appreciate the suggestion. The relevant hyperparameters (buffer length, unroll steps, penalty weight, LR scale) were chosen via task-specific hyperparameter search; in practice, buffer length and training unroll steps are mainly limited by GPU memory. Performance vs. test-time iteration counts and runtime/latency are detailed in the paper and Appendix A, and therefore in our opinion provide a full picture.
>
> While not grouped under a dedicated “ablation” section, several of these factors are already evaluated: Sec. 5.1.1 studies buffer-length variations, and Sec. 5.1.2 examines the stability and consistency of our method across varying problem setups. However, we agree that a more systematic presentation (e.g., grids/heatmaps) is a great idea, and will supplement this in the revision.
>
> Inference-Time Behavior
>
> We thank the reviewer for raising this important point. At test time, L-SR1 does not rely on any additional guardrails, damping, or fallback mechanisms. The learned projection guarantees that the preconditioner is positive semi-definite by construction, and the secant constraint encourages the update directions to approximate the true inverse Hessian.
>
> The interaction between the buffer, base matrix, and generated vectors during inference is fully captured by the pseudo-code in Algorithm 1, which accurately reflects the test-time procedure. Furthermore, Sec. 5.1.1 compares the directions produced by L-SR1 to the true Newton directions, showing that the learned projection consistently generates better-aligned, more Newton-like directions than a variant without the projection. Together with the runtime and test-time iteration data in Appendix A, this provides a complete picture of L-SR1’s inference behavior.
>
> 2. Addressing the Reviewer's Questions:
>
> Q1: Consistency checks or safeguard mechanisms?
>
> As discussed earlier, no additional safeguards are applied at test time. The learned projection ensures PSD preconditioners, and secant consistency guides Newton-like directions.
>
> Q2: Performance profile under matched iteration/clock budget
>
> All optimizers in Sec. 5.1.2 use the same update-step budget, ensuring fair comparison. Wall-clock runtime is reported in Appendix A. We will include curves showing the number of steps required to reach target errors in the supplementary material in the revision.
>
> Q3: Grid/heatmap of hyperparameters
>
> We agree this is a great idea. We will supplement the revision with a systematic grid/heatmap over buffer length, unroll steps, projection penalty, and learning-rate scaling, reporting final error, steps to target, and variance across seeds.

---

### Official Review · Reviewer_tDyX · 2025-10-28

**Soundness:** 2
**Presentation:** 2
**Contribution:** 1
**Rating:** 2
**Confidence:** 3

**Summary:**

The paper proposes L-SR1, a learned second-order optimizer that augments classical SR1 with a trainable low-rank preconditioner and a learned projection that enforces both the secant constraint and positive semidefiniteness, then applies it to analytic functions and monocular HMR.

**Strengths:**

- Principled design bridging QN and learning. The method is explicitly grounded in the QN update​, the secant condition, and the need for PSD preconditioners for descent directions; the learned projection aims to satisfy both simultaneously.


- Lightweight, limited-memory, dimension-invariant formulation. L-SR1 uses rank-one outer products with a fixed-size buffer and element-wise modules to generalize across problem sizes without retraining.


- Learned projection and per-coordinate step sizes. The optimizer integrates a vector generator and per-coordinate learning-rate generator; the projection improves convergence while keeping the model compact.

**Weaknesses:**

- **Insufficient Comparative Analysis**: The paper should compare against a wider set of HMR methods (both optimization-based and modern learned/learnable refiners beyond LGD/SPIN) and report more metrics (e.g., MPJPE, PA-MPJPE, PVE, jitter/contact, temporal stability) on more datasets (e.g., Human3.6M, EHF, AGORA) under matched settings. As written, the HMR main table includes only a few baselines. Moreover, the paper fails to provide an analysis of the computational cost and runtime of the proposed method versus more competitors beyond LGD. This essential data is needed to properly highlight the claimed efficiency.

- **Poor Visual Quality**: A major concern is the poor visual alignment of the fitted SMPL models in Figure 5 and Figure 9. For example, the case in the third row of Fig. 9 is clearly misaligned with the input image in the leg area. The meshes frequently appear misaligned with the subjects in the original images, suggesting a lack of robustness.

**Questions:**

- Why was HMR selected as the experimental testbed, and which limitations of existing HMR approaches does this work seek to overcome? The connections between the proposed method and the HMR are unclear. Please elaborate with stronger justification.

- Justification of Optimization: The fundamental advantage of optimization-based methods is often their ability to achieve higher accuracy by iteratively refining the fit to the image evidence. However, given that your method's performance (PA-MPJPE: 51.58) lags significantly behind current state-of-the-art learning-based methods (e.g., PromptHMR [CVPR 2025]: 36.6; CameraHMR [3DV 2025]: 35.1), the following question arises: If the accuracy of your optimization-based method is not superior to, and in fact is substantially worse than, its learning-based counterparts, what is the practical and methodological significance of using a much slower optimization approach?

- Scope of “no annotations / no fine-tuning.” For HMR, please specify the exact inner objective (2D keypoints? camera model? masks/contours?), any external detectors used, and how this complies with the “no annotations” claim; otherwise, the “10% data surpasses full training” result is hard to interpret.

- Supervision boundary. Clarify “no annotated data / no fine-tuning”: what signals supervise HMR (e.g., only 2D keypoints?), what detector(s) and confidence treatment are used, and how is the 10% data experiment constructed (splits, metrics, variance)?

- Discuss failure cases (e.g., heavy occlusion, noisy 2D keypoints input, temporal consistency).

---

> ### Author Response · Authors · 2025-11-20
> **Rebuttal**
>
> Dear Reviewer tDyX,
> Thank you for the careful and detailed review. We appreciate your feedback and address your concerns below. In several cases, the issues stem from insufficient clarity in the original manuscript regarding the scope and purpose of our method.
>
> 1. Main Concern: Comparative Analysis
>
> The primary goal of our paper is L-SR1 as a general learned second-order optimizer, not a new HMR architecture. The HMR fitting experiments are intended as a challenging benchmark for optimizer behavior, not to achieve SOTA HMR performance.
>
> We compare L-SR1 with:
> LGD, a learned optimizer trained under the same meta-training setup on AMASS, providing a direct evaluation of learned curvature handling;
> L-BFGS, a classical optimizer with no learned parameters, serving as a baseline for classical quasi-Newton behavior;
> an Adam, which was dropped eventually due to worse performance.
> These comparisons isolate the contribution of L-SR1’s learned projection and low-rank preconditioner under identical inner objectives (2D reprojection loss) and initialization. End-to-end HMR regressors rely on large-scale image-level 3D supervision and dataset-specific finetuning; thus, direct comparisons would be misleading. We note that modern SOTA HMR regressors could be added to Table 1 for reference only, without implying a meaningful comparison.
>
> Regarding computational cost, L-SR1 is efficient:
> Compared to LGD, it requires less runtime and memory on HMR tasks (Tables 2 and 3).
> Compared to L-BFGS (analytic experiments in App. A), it achieves substantially better convergence with minimal additional cost.
>
> 2. Adressing the Reviewer's Questions:
>
> Q1: Why HMR was selected as a testbed -
>
> HMR provides an interesting and challenging task where optimization historically played a significant role in fitting 3D human meshes. It is particularly suited for evaluating learned optimizers because it combines high-dimensional, non-convex objectives with noisy 2D observations. The learned optimization baseline LGD was previously introduced on this task, making it a natural choice to integrate and evaluate our L-SR1 optimizer under the same setting.
>
> Q2: Justification of optimization despite lower accuracy vs SOTA -
>
> L-SR1 does not aim to outperform fully supervised HMR regressors, which rely on large-scale 3D supervision and dataset-specific finetuning. Instead, it performs test-time optimization only with a 2D reprojection loss. The HMR experiments demonstrate methodological improvements—faster convergence, greater stability, and robustness—relative to other learned and classical optimizers under identical objectives.
>
> Q3–Q4: Supervision -
>
> During meta-training, L-SR1 uses 3D SMPL parameters from the AMASS dataset. The meta-training framework is identical to LGD: 3D SMPL parameters are used to generate 3D keypoints, which are then randomly projected to 2D; keypoint confidences are sampled from a Bernoulli distribution. The meta-loss obtaines supervision from the SMPL params, as well as 3D and 2D keypoint.
> At test-time fitting on 3DPW, the optimizer relies solely on the 2D keypoints which are given part of the 3DPW dataset, (obtained using OpenPose, as elaborated in the paper (App. C.2.3)). No 3D annotations or SMPL parameters are used at test time.
>
> The “10% experiment” uses only 10% of AMASS samples for meta-training, while evaluation is performed on the 3DPW test set as in the main experiments.
>
> Q5: Failure cases -
>
> Known limitations include heavy occlusions, noisy 2D keypoints, and per-frame temporal inconsistencies. These affect all optimization-based HMR methods using the same 2D loss and are not specific to L-SR1. We will clarify these points in the revision.
>
> Remark on Visual Quality:
>
> We appreciate the observation regarding some misaligned meshes. Quantitatively, L-SR1 consistently outperforms SR1, L-BFGS, and LGD in reprojection loss and convergence speed. In the revision, we will update the figures to better reflect overall performance, including both successes and representative failures.

---

> > ### Comment · Reviewer_tDyX · 2025-11-27
> >
> > I appreciate the authors’ clarification regarding the use of HMR as a testbed. However, I still have significant concerns about its weak performance. My assessment remains inclined toward rejection.

---

### Official Review · Reviewer_kNdW · 2025-10-28

**Soundness:** 3
**Presentation:** 3
**Contribution:** 3
**Rating:** 6
**Confidence:** 3

**Summary:**

This work introduces ​​L-SR1 (Learned Symmetric-Rank-One)​​, a novel learned second-order optimizer that enhances the classical SR1 algorithm by integrating a lightweight, trainable preconditioning unit. This unit generates data-driven vectors to construct positive semi-definite rank-one matrices, with a key innovation being a learned projection mechanism that enforces the critical secant condition.

**Strengths:**

The theoretical foundation of this work is  presented with clarity.

**Weaknesses:**

The method is only tested on HMR, a specific application in image processing.
It is not know whether this method is applicable or not for other tasks.

**Questions:**

1. As a reviewer not deeply familiar with optimizers, I am unclear about the specific contribution of this work in the optimization field. While the authors propose an improved SR1 method, I am uncertain whether such approaches are common or novel in optimization research.
2. Ignoring novelty, the work is valuable as it applies learned optimizers to SR1. The authors appear to address key challenges in adapting SR1 into a learned framework.
3. I have some concerns regarding the sufficiency of the experimental validation. The evaluation is primarily centered on the Monocular Human Mesh Recovery (HMR) task. I am curious to know in which other tasks or domains this method could be effectively applied. HMR itself can be addressed by both optimization-based methods (e.g., SMPLify) and regression-based approaches. This leads to a question of whether the proposed method is specifically tailored to similar iterative optimization frameworks. If so, its potential application domain might be somewhat limited, though I acknowledge this could be a general challenge for the field of learned optimizers rather than a specific shortcoming of this paper.
4. Furthermore, the comparative analysis in the experiments appears relatively limited. For instance, Table 1 primarily reports results on the 3DPW dataset. It would strengthen the paper to include comparisons on other standard benchmarks in this domain. Additionally, the baseline methods used for comparison seem to be from 2020 or earlier. Including comparisons with more recent state-of-the-art approaches would provide a more compelling and up-to-date demonstration of the method's competitiveness.

---

> ### Author Response · Authors · 2025-11-20
> **Rebuttal**
>
> We thank the reviewer for the careful reading and constructive feedback. We address the concerns raised and provide answers to the specific questions posed.
>
> 1. Addressing the Reviewer's Concerns - Novelty and Broader Applicability
>
> L-SR1 introduces a learned extension of the classical SR1 quasi-Newton method, combining a trainable low-rank preconditioning unit with a learned projection that enforces positive semi-definiteness and the secant constraint. This ensures stable, Newton-like update directions (Sec. 5.1.1). While SR1 itself is classical, integrating these principles into a lightweight, learned optimizer is, to our knowledge, novel and advances the field by bridging classical second-order optimization with task-agnostic learned optimizers.
>
> Although HMR is the main real-world testbed, L-SR1 is not tailored specifically to this task. Its generalization is demonstrated through analytic experiments on quadratic and non-convex functions, and through dimension-agnostic behavior, handling varying problem sizes without retraining. HMR provides a challenging, high-dimensional benchmark, but L-SR1’s design allows it to be applied to any iterative gradient-based optimization problem, illustrating broader applicability beyond the experiments reported.
>
> 2. Addressing Further Questions:
>
> “HMR itself can be addressed by both optimization-based methods (e.g., SMPLify) and regression-based approaches. This leads to a question of whether the proposed method is specifically tailored to similar iterative optimization frameworks.”
>
> Response: HMR provides a challenging, high-dimensional, real-world testbed that is well-suited to evaluate optimizer generalization. While L-SR1 is not specialized for HMR, this task allows us to test the optimizer under realistic conditions with high-dimensional inputs, multiple local minima, and real-world data variability. The goal is to assess optimizer performance and generalization, rather than achieving SOTA HMR accuracy.
>
> “It would strengthen the paper to include comparisons on other standard benchmarks in this domain. Additionally, the baseline methods used for comparison seem to be from 2020 or earlier.”
>
> Response: To the best of our knowledge, optimization-based HMR solvers have largely been replaced by large, end-to-end regressors that are heavily fine-tuned. In contrast, our method is trained on AMASS with synthetically generated 2D keypoints and evaluated directly on real 3DPW data without explicit fine-tuning. In our evaluation, we compare L-SR1 to both classical optimization and learned-optimization baselines, which are the relevant frameworks for evaluating optimizer performance. While we could add recent SOTA HMR methods as a reference for context, a direct comparison would not be apples-to-apples due to the fundamentally different training regimes and objectives.

---

### Official Review · Reviewer_zsPf · 2025-10-29

**Soundness:** 3
**Presentation:** 2
**Contribution:** 2
**Rating:** 4
**Confidence:** 4

**Summary:**

This paper presents a learned second-order optimizer, named Learned-SR1 (L-SR1), which proposes to use a trainable preconditioning unit to enhance the classical Symmetric-Rank-One (SR1) algorithm while ensuring positive definite preconditioning matrices that satisfy the
secant constraint. The proposed method shows superiority over existing optimizers in benchmark tasks such as Human Mesh Recovery (HMR), and also demonstrates the highest performance profile and converging to the global minimum faster.

**Strengths:**

A lightweight, self-supervised learned optimizer that integrates a trainable preconditioning unit into the SR1 framework is introduced.

A learned projection mechanism that enforces both the secant condition and positive semi-definiteness, preserving core Quasi-Newton properties within a learned architecture is introduced.

Experiments show that the proposed method works well on HMR.

The paper is well written and easy to read.

**Weaknesses:**

The claimed generalization of the proposed Learned-SR1 is not effectively validated. Currently, there is only simple evaluation on HMR task, and the compared baselines do not represent the current state-of-the-art for HMR.

The paper lacks comparison with more optimization algorithms, e.g., AdamW, AdaHessian, etc. Moreover, the theretical analysis of the learned projection mechanism is insufficient.

Currently, the evaluation is conducted only on a single dataset (3DPW), which fails to demonstrate the effectiveness and robustness of the proposed method.

This work lacks a thorough ablation study for each design in the proposed method.

**Questions:**

How does L-SR1 perform on tasks beyond HMR?

There is no visual comparison for HMR. It would be better if visual comparisons could be included in Figure 5.

---

> ### Author Response · Authors · 2025-11-20
> **Rebuttal**
>
> We thank the reviewer for the careful reading and constructive feedback. We address the concerns raised and provide answers to the specific questions posed.
>
> 1. Addressing the Reviewer's Concerns:
>
> Evaluation and Generalization
>
> The robustness, generalization, and effectiveness of L-SR1 are demonstrated through both analytic experiments and the real-world HMR task. While our goal is not to achieve SOTA end-to-end HMR performance—which relies on large-scale supervised training and specialized architectures—HMR provides a challenging real-world optimization benchmark where L-SR1 demonstrates strong iterative refinement performance relative to other optimization-based methods.
> Generalization is shown primarily through training on AMASS using a framework where input 2D data is generated synthetically from SMPL parameters, and then evaluating on 3DPW, which contains real-world images with different pose distributions and detection noise. L-SR1 also generalizes across varying problem dimensions without retraining, as confirmed in the analytic experiments.
>
> Comparison with other optimizers
>
> Sections 5.1.1 and 5.1.2 include comparisons with classical and learned optimizers, including L-BFGS, Adam, and AdaHessian. These experiments allow us to systematically assess convergence behavior, performance profiles, and the impact of key components of L-SR1. We note that we also evaluated both Adam and AdamW during development and L-SR1 consistently outperformed them. However, because AdamW’s main advantage arises in settings where weight decay plays a central role—an aspect not relevant in our optimization-centric setup—we chose not to include AdamW in the paper for conciseness.
>
> Theoretical analysis of the Learned Projection
>
> The learned projection mechanism generates PSD matrices by construction, while the secant constraint encourages these matrices to approximate the true inverse Hessian directions. This property is explicitly validated in Sec. 5.1.1 (Quadratic Functions), where directions produced with the learned projection tend to align with Newton directions, whereas removing the projection leads to deviations. These experiments provide both practical and conceptual insight into the effect of the learned projection.
>
> Ablation studies
>
> Analytic experiments (Sec. 5.1) effectively validate the contribution of the learned projection mechanism and study the influence of buffer size, serving as targeted ablation-style analyses. Further ablations supporting other design choices are presented in Appendix B. The ablations section will be expanded in the revision.
>
> 2. Addressing the Reviewer's Questions:
>
> Q1: How does L-SR1 perform on tasks beyond HMR?
> L-SR1 is extensively evaluated on analytic optimization tasks (Sec. 5.1), including quadratic functions, Rosenbrock, and other standard objective families. On these tasks, it demonstrates faster convergence, higher stability, and stronger performance profiles than classical optimizers such as SR1, L-BFGS, and Adam. These experiments show that L-SR1’s learned updates generalize beyond HMR to a broad set of optimization problems.
>
> Q2: No visual comparison for HMR — can it be added?
> A visual comparison between L-SR1 and LGD is provided in Appendix E (Fig. 9). We will ensure this figure is clearly referenced in the main text in the revision to highlight the qualitative differences in convergence behavior.

---

### Meta-Review · Area_Chair_MXMt · 2026-01-20

**Summary:**

This paper proposes L-SR1, a lightweight learned second-order optimizer that augments the classical SR1 method with a trainable low-rank preconditioner and a learned projection enforcing secant consistency and positive semi-definiteness.

- Reviewer zsPf's concern is the limited experimental validation and lack of comparisons with stronger and more diverse baselines.

- Reviewer kNdW questions the clarity and breadth of the contribution from an optimization perspective.

- Reviewer tDyX criticizes the paper’s limited comparisons, poor HMR visual results, unclear supervision claims, and lack of justification for the optimization approach, given low accuracy relative to state-of-the-art methods.

- Reviewer n9Pd emphasizes the need for broader comparisons, deeper ablations, and clearer inference details.

**Reviewer Concerns:**

Overall, the authors have addressed the main concerns raised by three of the reviewers. However, they have not resolved Reviewer tDyX’s issues, particularly regarding the weak empirical performance and its justification relative to state-of-the-art methods.

**Reviewer Scores:**

After the rebuttal, the authors were still unable to address Reviewer tDyX’s concerns, and the reviewer’s score remained unchanged. The paper would benefit from broader validation, stronger baseline comparisons, and clearer justification of the optimizer’s practical value.

---

### Decision · Program_Chairs · 2026-01-26

Reject